# Multiple Antitumor Molecular Mechanisms Are Activated by a Fully Synthetic and Stabilized Pharmaceutical Product Delivering the Active Compound Sulforaphane (SFX-01) in Preclinical Model of Human Glioblastoma

**DOI:** 10.3390/ph14111082

**Published:** 2021-10-26

**Authors:** Alessandro Colapietro, Alessandra Rossetti, Andrea Mancini, Stefano Martellucci, Giuseppe Ocone, Fanny Pulcini, Leda Biordi, Loredana Cristiano, Vincenzo Mattei, Simona Delle Monache, Francesco Marampon, Giovanni Luca Gravina, Claudio Festuccia

**Affiliations:** 1Laboratory of Radiobiology, Department of Biotechnological and Applied Clinical Sciences, University of L’Aquila, 67100 L’Aquila, Italy; alecolapietro@gmail.com (A.C.); alessandra.rossetti@univaq.it (A.R.); mancio_1982@hotmail.com (A.M.); giuseppe.ocone@student.univaq.it (G.O.); giovanniluca.gravina@univaq.it (G.L.G.); 2Biomedicine and Advanced Technologies Rieti Center, Sabina Universitas, 02100 Rieti, Italy; s.martellucci@sabinauniversitas.it (S.M.); v.mattei@sabinauniversitas.it.it (V.M.); 3Laboratory of Vascular Biology and Stem Cells, Department of Biotechnological and Applied Clinical Sciences, University of L’Aquila, 67100 L’Aquila, Italy; fanny.pulcini@graduate.univaq.it (F.P.); simona.dellemonache@univaq.it (S.D.M.); 4Laboratory of Medical Oncology, Department of Biotechnological and Applied Clinical Sciences, University of L’Aquila, 67100 L’Aquila, Italy; leda.biordi@univaq.it; 5Department of Clinical Medicine, Public Health, Division of Human Anatomy, University of L’Aquila, 67100 L’Aquila, Italy; loredana.cristiano@univaq.it; 6Department of Radiological, Oncological and Pathological Sciences, La Sapienza University of Rome, 00185 Rome, Italy; francesco.marampon@uniroma1.it; 7Department of Biotechnological and Applied Clinical Sciences, Division of Radiotherapy, University of L’Aquila, 67100 L’Aquila, Italy

**Keywords:** GBM, sulforaphane, SFX-01, hypoxia, epithelial-to-mesenchymal trans-differentiation (EMT), stemness and tumor recurrence

## Abstract

Frequent relapses and therapeutic resistance make the management of glioblastoma (GBM, grade IV glioma), extremely difficult. Therefore, it is necessary to develop new pharmacological compounds to be used as a single treatment or in combination with current therapies in order to improve their effectiveness and reduce cytotoxicity for non-tumor cells. SFX-01 is a fully synthetic and stabilized pharmaceutical product containing the α-cyclodextrin that delivers the active compound 1-isothiocyanato-4-methyl-sulfinylbutane (SFN) and maintains biological activities of SFN. In this study, we verified whether SFX-01 was active in GBM preclinical models. Our data demonstrate that SFX-01 reduced cell proliferation and increased cell death in GBM cell lines and patient-derived glioma initiating cells (GICs) with a stem cell phenotype. The antiproliferative effects of SFX-01 were associated with a reduction in the stemness of GICs and reversion of neural-to-mesenchymal trans-differentiation (PMT) closely related to epithelial-to-mesenchymal trans-differentiation (EMT) of epithelial tumors. Commonly, PMT reversion decreases the invasive capacity of tumor cells and increases the sensitivity to pharmacological and instrumental therapies. SFX-01 induced caspase-dependent apoptosis, through both mitochondrion-mediated intrinsic and death-receptor-associated extrinsic pathways. Here, we demonstrate the involvement of reactive oxygen species (ROS) through mediating the reduction in the activity of essential molecular pathways, such as PI3K/Akt/mTOR, ERK, and STAT-3. SFX-01 also reduced the in vivo tumor growth of subcutaneous xenografts and increased the disease-free survival (DFS) and overall survival (OS), when tested in orthotopic intracranial GBM models. These effects were associated with reduced expression of HIF1α which, in turn, down-regulates neo-angiogenesis. So, SFX-01 may have potent anti-glioma effects, regulating important aspects of the biology of this neoplasia, such as hypoxia, stemness, and EMT reversion, which are commonly activated in this neoplasia and are responsible for therapeutic resistance and glioma recurrence. SFX-01 deserves to be considered as an emerging anticancer agent for the treatment of GBM. The possible radio- and chemo sensitization potential of SFX-01 should also be evaluated in further preclinical and clinical studies.

## 1. Introduction

High-grade gliomas (HGGs) are primary brain tumors in adults [1,2]. Classically, the World Health Organization (WHO) has histologically classified them as, anaplastic astrocytomas (AA), oligodendrogliomas (AOAs), and grade IV glioblastomas (GBM) [2]. The last account for about 60% of HGGs and 50% of all malignant brain tumors and present the worst prognosis. The current standard of care (SOC) for GBMs involves a multimodal approach, including surgery, followed by radiation and chemotherapy [3]. Surgical resection is performed with the goal of achieving a “maximal safe resection”, and, in this way, is used to relieve mass effect and set the stage for multimodal adjunctive therapy. Unfortunately, this surgical goal is not always achievable. The efficacy of chemotherapy has been demonstrated in improving the patient’s prognosis Temozolomide (TMZ) has sufficient cerebral penetration; however, its efficacy is limited to patients with methylation of the gene for the DNA repair enzyme O6-methylguanine-DNA methyltransferase (MGMT). These patients may also develop resistance to the drug and recurrence of the disease through the involvement of different biochemical mechanisms [4]. Landmark studies by Stupp et al., demonstrated improved 2-year survival (from 10.9% to 27.2%) when adding EBRT to TMZ. HGGs cells survive, differentiate, and grow well in hypoxic niches [5,6,7], which upregulate the expression of different molecules including transcription factors, cytokines, and chemokines.

Hypoxia-inducible factors (HIF-1a and HIF2a) [8] are potent inducers of vascular endothelial growth factor (VEGF), a transcription factor that plays a key role in promoting vasculogenesis (neoangiogenesis coupled to vascular mimicry). Indeed, HGGs tissues show excessive vascularization, which jeopardizes the blood–brain barrier and promotes further damage, edema, and necrosis. The latter is also able to maintain hypoxia [8]. Histologically, GBMs develop a typical morphological type of necrosis, called “palisading necrosis” [9] consisting of small, irregular regions of necrosis surrounded by dense accumulations of tumor cells.

EBRT and temozolomide-based chemotherapy essentially exert cytotoxic effects by damaging the DNA and inducing DNA breaks. Limitations exist with these treatments when they are used as single therapeutic modalities, due to the high cell heterogeneity of tumors and the deregulation of several cell signaling cascades. Resistant cells develop due to the disappearance of DNA breaks due to increased DNA repair. Multimodal treatments targeting distinct molecular pathways are an effective strategy in GBM [10]. Such strategies can circumvent some of the obstacles to individual treatments.

Genetically, the tumor heterogeneity of GBM is represented by four clinically relevant subtypes [11]: proneural (PN), neural (NL), classical (CL), and mesenchymal (Mes) also if extreme heterogeneity exists even within each subgroup. This classification has strong implications in the selection of anti-target therapies [12]. The MES subtype represents the most aggressive phenotype, which is strongly associated with poor prognosis, compared with the PN subtype. A shift from the PN to MES subtype can occur in patients following radiation therapy and chemotherapy and in those who develop recurrence [13]. Therefore, a reversion of this transition may serve as a useful therapeutic strategy [14].

Similar to all aggressive solid tumors, GBM contains high percentages of transformed, self-sufficient, proliferating, and multipotent cells that maintain the phenotype (marker expression) of stem cells. These cells are also called glioma initiating cells (GIC) and express GSC markers, such as CD133, Stro1, Oct3/4, Sox2, and Nanog. These cells can express different levels of MES markers, including CD44, CD90, CD105, vimentin and N-cadherin, depending on their different genetic subtype membership. GICs differ from GSCs in their greater ability to proliferate in vitro which allows them to reconstitute the tumor when the therapy has taken effect and reduced tumor mass. GSCs, on the other hand, have very low proliferative rates which, due to their poor replicative capacity, can be protected by drugs and therapies, constituting the totipotent reservoir from which to restart after relapse to therapy. Normally GIC cells develop from GSC cells which are stimulated to proliferate. Stem cell markers are usually also mediators or creators of one or the other stem cell attribute while stemness refers to a wide range of attributes, some of which are completely unrelated to other, which the elimination of one can leave unchanged. This cell population can develop mainly in the presence of palisade necrosis with high inflammation [15,16,17,18]. The hypoxic areas are, in fact, enriched with GSC but these cells can also be found in non-hypoxic areas. Considering the higher proliferation rates of GICs compared to GSCs, it is important to consider the possible interconversion between GSCs into GICs and between GICs and differentiated cells and vice versa. This phenomenon has been well documented in gliomas and underlies the cellular plasticity of this neoplasm. [19,20]. Infiltrating GICs result in lower sensitivity to therapies and can lead to tumor recurrence in most patients [16]. Moreover, GBM cells are typically surrounded by resident non-tumor cells, including infiltrating lymphocytes and glioma-associated microglia/macrophages constituting the tumor stroma, which are responsible for the high malignancy of GBMs. This cell population works in collaboration with tumor cells and endothelial precursors to support angiogenesis, growth, local invasiveness [16], and sensitivity/resistance to therapies [21].

In recent years, interest in the chemo preventive and therapeutic effects of botanical compounds [22,23,24,25,26,27] has remarkably increased. Sulforaphane (SFN) is a natural compound belonging to the isothiocyanates. It is present in broccoli sprouts, as well as other cruciferous vegetables including cabbage and kale [24]. This is commonly available over the counter, without a prescription, in many jurisdictions. It is also popularly used by cancer patients.

Although SFN can significantly engage the antioxidant activity of nuclear factor erythroid 2-related factor 2 (NRF2) [28,29] and protect normal cells to toxic agents and therapies such as radiations, several lines of experimental evidence have demonstrated that SFN shows cytotoxic and pro-apoptotic effects in different cancer cell types including glioma cells [23,24,25,26,27,28,29].

Moreover, an altered redox homeostasis is observed in GBMs, and this triggers the activation of various survival pathways leading to disease progression and chemo resistance [24,30,31,32,33]. It is necessary to consider, however, that changes in the ROS threshold by antioxidant compounds may differently affect the viability of tumor cells; for example, if we consider the literature on the biological effects of heme-oxygenase-1 (HO-1), we can observe that its upregulation can promote cell cycle arrest and cellular death in some tumors, whereas it has also been associated with tumor survival and progression in other neoplasms [34,35,36]. Moreover, the antitumor effects of SFN are strongly related to inhibition of multiple signaling pathway able to induce autophagy and/or apoptosis with further alteration of the oxidative stress [36]. For example, SFN is able to interfere with STAT3 [37,38], sonic hedgehog pathway [39], Hsp90 [40], histone deacetylase (HDAC) [41], and other signaling pathways.

To improve the stability of this compound, a fully synthetic SFN (1-isothiocyanato-4-methylsulfinylbutane), were developed. SFN was complexed with α-cyclodextrin) and registered from Evgen pharmaceuticals PLC as SFX-01 (Figure 1) under the European patent EP2854861 [42]. SFX-101 contains a racemic mixture of synthetic D and L isoforms of SFN. The natural enantiomer is the L form [42]. Literature reports have demonstrated that there is no difference in potency between the enantiomers, but the racemate seems to be more potent in some biological systems. The biological activity of the active substance is conferred by the SFN moiety. The alpha-cyclodextrin is essential for maintaining the stability of the SFN. This component is biologically inert and ineffective in several cell systems in vitro including GBM cells [39,43].

In addition, SFX-01 is easily absorbed in the digestive tract indicating that the presence of cyclodextrin favors the bioavailability of SFN. In literature there is a sole open-label phase II clinical trial which was started in patients with ER+ metastatic breast cancer resistant to hormonal therapies (ClinicalTrials.gov Identifier: NCT02970682) [44]. Patients were enrolled into one of three study arms, in order to demonstrate the safety and efficacy of SFX-01 when used in combination with aromatase inhibitors, tamoxifen, and fulvestrant. Results of this study, started in 2016 with completion in the January 2019. Results are published in date 29 January 2020 [45] and as personal communication to ESMO congress 2019 [46]. The study demonstrated that SFX-01 300 mg BID was safe and well tolerated in patients with ERþ and HER2-mBC. SFX-01 in combination with ET demonstrated anti-tumor activity and prolonged disease stabilization in pre-treated patients who were progressing on the background ET at study entry.

Here, we observed that SFX-01 reduced GBM cell growth, as well as the formation and growth of glioma spheres at similar micro molar concentrations of SFN. In addition, SFX-01 increased the cell differentiation of GBM cell lines and GICs. Biochemical analyses showed that cell death requires the induction of both intrinsic and extrinsic apoptotic pathways. We confirmed published data indicating the down regulation of PI3K/Akt/mTOR pathways, as well as STAT3 and HDAC activity. We also demonstrated that SFX-01 strongly reduced tumor growth in subcutaneously injected mice, as well as increased disease-free survival and overall survival in orthotopic intracranial GBM models, supporting a role in glioma treatment as a single therapy. Another manuscript describing the sensitization effects of SFX-01 versus RT is in preparation.

## 2. Results

### 2.1. SFX-01 Inhibited Proliferation of GBM Cells and GICs In Vitro

First, we tested the antiproliferative effects of SFX-01 administered at 2.5, 5, 10, 20, and 40 µM on GBM and patient-derived GIC lines, as indicated in the Material and Methods section. GBM cell viability was assessed at 12, 24, 48, 72, and 96 h, using a Cell Counting Kit-8 (CCK-8; Dojindo Molecular Technologies Inc., Tokyo, Japan). The proliferation rate was calculated considering the percentage values observed at the experimental points with respect to controls (100%). Cyclodextrin is inert in culture test and showed no effects on several tumor cells including GBM cells as indicated by previous data from E. Leung [43] at the Auckland Cancer Society Research Centre, Medical Science, Faculty of Medical and Health Sciences (Auckland, New Zealand) and similar to those observed from Simoes et al. [39]. In Figure 2A, we show the inhibitory effects of SFX-01 in U251, U87MG, and T98G cells. These experiments demonstrate that SFX-01 reduced cell proliferation and viability in a dose- and time-dependent manner, and that lower doses of SFX-01 may be administered to cultures for a longer time and still have significant effects. Morphological changes were observed after the administration of SFX-01, with the acquisition of cellular extensions associated with a probable more-differentiated astrocyte phenotype (Figure 2B). This morphology was more evident at doses of SFX-01 > 10 µM and after 48 h of culture. The IC50 values were calculated using the Grafit software, and, in Figure 2C, we show the values obtained for six GBM cell lines. The IC50s ranged between 9.2 (U87MG) and 22.33 (T98G) µM. Next, we tested the effects of SFX-01 in GIC cultures by treating pre-formed neurospheres (experiment type 1) with different doses of the compound for 1 week. In Figure 2D, we show that SFX-01 reduced the size and the number of GSCs5 neurospheres in a dose-dependent manner. The IC50 values were calculated, considering the percentage of growth under experimental conditions versus the untreated controls.

For this analysis, we used three GIC cultures and found IC50s of 8.1 µM (BT48EF), 12.7 µM (BT12M), and 25.5 µM (GSCs-5), as shown in Figure 2E. Next, in order to determine the effects of SFX-01 on cell proliferation, we verified the expression of Ki67, by FACS, in the controls or after the administration of SFX-01 at 5, 10, and 20 µM in four GBM (U251, U87MG, T98G, and A172) cell lines and two GIC (BT12M and GSGs-5) cultures (Figure 3). We demonstrated that SFX-01 reduced the Ki67 percentage in a dose-dependent manner and, thus, the cell proliferation in both GBM (Figure 3A) and GIC (Figure 3C) cell cultures. Next, we analyzed the effects of SFX-01 on the formation of neurospheres starting from single cells (experiment 2: stemness experience). In Figure 3B, we demonstrate that SFX-01 reduced the stemness of GSGs-5 in a dose-dependent manner, indicating its role in the development of GICs. In Appendix A, we show the representative FACS profiles of cell lines treated with SFX-01 at 10 and 20 µM and of the control U87MG cell line. The percentages of cells in different cell cycle phases demonstrated an increment in cells in G2M and S, suggesting a block of progression in these cell cycle phases. FACS analyses were also repeated for U251 and T98G cells, with similar results.

Appendix A shows the changes in the U87MG, U251, and T98G cell percentages in the different cell cycle phases over time, demonstrating a time-dependent accumulation of cells in the G2/M phase, with a sensitive time-dependent reduction of the S phase.

Western blot analysis (Appendix A), performed on the U87MG cell line, showed an increased expression of cyclin B1, p27, and p21, and a reduced expression of cyclin D1 and cdk4 with 5 and 10 µM SFX-01 administration, supporting arrest in the G2/M and S phases of the cell cycle.

### 2.2. SFX-01 Reduces Stemness and Malignancy in GBM Cell Lines

Pro-neural to mesenchymal transition (PMT) of glioma is known to be associated with aggressive phenotypes, unfavorable prognosis, and treatment resistance [47]. This process is analogous to epithelial-to-mesenchymal transition (EMT) of other solid tumors. The acquisition of PNT phenotype provides the phenotypic plasticity necessary for the multi-therapy resistance. We hypothesize that SFX-01 could have pro-differentiating effects with reduction of malignancy. So, we took Oct3/4 into consideration as an example for stem cell markers, CD44, CD90, and Stro1 for MES markers; β3-tubulin, neurofilaments (NF200), and nestin for neural markers; and GFAP as an astrocyte marker by using established glioma (U251, U87MG, A172 and T98G) cell lines and patient-derived GICs (BT12M). In order to maintain the differentiated phenotype of established cell lines we cultured the cells in serum containing culture medium. Differently, to maintain a less-differentiated/undifferentiated status for GICs, these cultures were treated in serum-free culture medium (see materials and methods); however, it is necessary to consider that the presence of serum alone allows for the partial differentiation of GICs [17]. In Figure 4A, we show western blots performed for U87MG cells showing the time-dependent downregulation of the Oct3/4 and CD44 markers, as well as the up-regulation of GFAP, β3-tubulin, and nestin. The expression of CD44, CD90, Stro-1, β3Tubulin, NFH (NF200), and GFAP in U87MG cells was analyzed also by FACS and data shown in Figure 4B. These effects were time dependent and required a dose of SFX-01 < 10 µM and 24–48 h to be markedly evident. The semi-quantitative analyses (normalized densitometric units) performed in different western blots of U87MG, U251, A172 and T98G glioma cell extracts (Figure 4C) support the idea of a common differentiated process induced by SFX-01 administration.

Accordingly with these data, we hypothesized that, by inducing a more differentiated phenotype, SFX-01 could modify the malignancy of glioma cell lines as the result of the reversion of PNT [48]. So, we decided to assay the malignancy of tumor cells in vitro by using two functional tests: (1) the matrigel invasion by using transwell chambers and (2) the vasculomimicry assay. Vasculogenic mimicry is associated with malignancy of a tumor and prognosis by increasing local invasion and vasculogenesis [49].

Glioma cells can be also induced to trans-differentiate in endothelial like cells or pericytes and participate to the vascular structure of glioma [50,51]. Figure 4D shows representative filters of transwell chambers in which U87MG cells were tested for the modulation of their migratory/invasive capabilities by 5 and 10 µM SFX-01 administration. We observed that SFX-01 had decreased the number of matrigel invading/invaded cells suggesting a less motile phenotype. This effect was evident also in U251. A172 and T98G glioma cells (Figure 4E). The inhibition of malignancy was also demonstrated through the changes of vasculomimicry ability. In Figure 4F we show the VM analyses performed in U87MG and U251 cell models. Also in this case, the administration of 5 and 10 µM SFX-01 reduced tube formation in both cells. The effects were analyzed by the analysis of the branching index by using the ImageJ software (version 1.8.0), as better specified in the Materials and Methods section.

In addition (as shown in the Appendix A), we found that SFX-01 induced changes in differentiated (GFAP, β3-tubulin, and NFH; NF200) and stem/MES (CD44, CD90, and Stro-1) markers in BT12M cells, a GIC cell line with a basal MES phenotype [52,53,54]. In Appendix A, we show confocal images demonstrating that GFAP expression increased when serum-free BT12M sphere cultures were cultured with 10 and 20 µM SFX-01. The spheres were disaggregated and analyzed with FACS for further marker quantification. In Appendix A, we show that the percentage of cells positive for stem/MES markers (CD44, CD90, and Stro-1) was decreased by SFX-01 administration. By contrast, the percentage of cells expressing β3-tubulin and NFH increased with SFX-01 administration. Moreover, SFX-01 could allow for the disaggregation of the spheres, inducing cell adhesion to the culture plastic in serum-free conditions. This phenomenon does not represent a real differentiation but suggests a trend towards differentiation. It is no coincidence that differentiated cultures of BT12M (Appendix A) showed high levels of GFAP, indicating type II astrocytes.

### 2.3. SFX-01 Reduced HIF1α Expression Downregulating Neoangiogenesis

In accordance with what was previously reported for SFN [52], we demonstrated that SFX-01 reduced the expression of HIF1α in glioma cells. In Appendix A, we found that cell extracts harvested from U87MG and U251 cells treated with 5 and 10 αM SFX-01 showed reduced HIF1α expression. We also analyzed the secretion of HIF1α and observed that conditioned medium from U87MG cells showed high levels of HIF1α and that SFX-01 reduced its expression (Appendix A). A similar trend was observed also for U251 cells. This is an important biological phenomenon since HIF1α increases stem cell recruitment and reduces cell differentiation and down-modulates angiogenesis. So, we analyzed whether the tubule formation of brain-derived endothelial cells (HBMVECs) was slowed down or not by the addition of conditioned media collected from untreated or SFX-01-treated U87MG and U251 cells (Appendix A). Tubule formation assays was analyzed after 15 h of culture. We observed that SFX-01 reduced glioma cell-mediated tubule formation, especially when the GBM cells were pre-treated for 24 h with 5 or 10 µM SFX-01. Parallelly, in order to verify if the above considered CMs reduced the proliferation of endothelial cells or were toxic, proliferation/cytotoxicity analyses were performed. We found that 15 h of treatment with SFX-01 as well as with CMs from glioma cells pre-treated or not with SFX-01 was ineffective for endothelial cells (Appendix A). However, although no direct cytotoxicity effects were observed when tested in culture, a variable proportion of endothelial cells that are inhibited to form tubes seemed not align adhering to the matrigel and remained in suspension. These cells could die as time progressed and cell death could be for anoikis. Pro-apoptotic effects could be a secondary mechanism to non-adhesion and tube formation. As conclusion of this first part of results we could state that SFX-01 may be very effective in eliminating or reducing GICs percentage and vasculogenesis.

### 2.4. Sulforaphane Induces Reactive Oxygen Species (ROS), Reduces Basal Autophagy, and Activates Apoptosis-Mediated Cell Death in GBM Cell Lines and GICs

In order to verify the drug modality of action (MOA) of SFX-01 and knowing that SFN increases NRF2 expression and NRF2-mediated detoxifying enzymes production, we verified whether SFX-01 affected ROS production and if this was associated with the antiproliferative effects shown by this compound. So, we tested U87MG and U251 glioma cells treated with 5 and 10 µM SFX-01 for the activation of the NRF2 pathways. We chose to use doses of SFX-01 lower than the IC50 and, more precisely, near the IC20 and IC10 values, to ensure that the effects were not toxic. Treatment was maintained 12–24 h when cell lysates were performed. At the same time cells were collected for ROS and MDA measurements. In Figure 5 we show the effects of SFX-01 at 12 h of treatment and maintained also for longer times. We demonstrated that, in agreement with literature data obtained with SFN, NRF2 levels were increased in a dose-dependent manner by SFX-01 administration while those of KEAP-1 (the inhibitor of the NRF2 pathways) resulted reduced. Nevertheless, the expression levels of detoxifying enzymes such as Sod1 and Sod2 were reduced in both glioma cell lines (Figure 5A). By contrast, HO-1 levels showed no changes in expression in U87MG, while it was strongly activated in U251 cells, also if at higher dose of 10 µM. Our data also demonstrated an increased expression levels of RAGE and p65-NFkB leading to ROS production. Reduction in Sirt-1 levels is also in agreement with the higher ROS production. So, in glioma cells the NRF2 pathways was not activated by SFX-01 administration. These data are contrasting with respect to the antioxidant role of NRF2. Notwithstanding, SFN may promote the accumulation of ROS and cause cell death in different cancer cell lines. Next, we quantified the ROS levels by measuring the immunofluorescence intensity from 2′,7′-dichlorofluorescein diacetate (DCFDA) production, as well as the levels of malondialdehyde peroxidation (MDA) by using specific enzymatic analyses. For this analysis, we tested the entire range of SFX-01 concentrations (covering the IC10, IC20, and IC50 values). In Figure 5B, we show that SFX-01 significantly increased the levels of MCFDA and MDA in the U87MG and U251 glioma cell lines. Several literature data, however, indicate that SFN sensitized cells to chemotherapeutics in different solid tumors [55,56] or radiation therapy [manuscript in preparation, [57] by enhancing ROS and apoptosis [58].

It has been demonstrated that increased ROS production induces DNA damage, which is immediately signaled through the phosphorylation of H2AX on serine139 (γ-H2Ax), the earliest indicator of DNA strand breaks (DSBs). This phosphorylated protein localizes near DNA strand breaks and recruits other proteins to the damaged site. The formation of DSBs may be detected by assessing the expression levels of γ-H2Ax in glioma cells by Western blotting, ELISA, and immunofluorescence. In Figure 5C, we show the Western blot analysis for γH2Ax expression in U87MG, A172, and U251 glioma cells cultured for 24 h with increasing doses of SFX-01. In these Western blots, it was clear that there was no dose-dependent increase, except for A172 cell line. Indeed, upon increasing the concentration, the other two lines behaved in a similar way, showing an increase first and then a decrease. On a purely representative basis, the immunofluorescence detection of γ-H2Ax foci is shown for U87MG cells in Figure 5D. The data confirmed that SFX-01 induced DSB production in a dose-dependent manner, as indicated by the increased γH2Ax expression in concerned nuclei.

ELISA determinations performed on nuclear extracts from untreated and treated cells confirmed that SFX-01 mediated DNA damage, reduced DNA repair and accumulated DSBs since γH2Ax levels did not completely return to baseline in the SFX-01-treated cells with the respect to RT treated cells used as control (Appendix A). In addition, the co-administration with the ROS-scavenger/e ROS inhibitor, N-acetyl-L-cysteine (3 mM, Appendix A) reduced the γH2Ax levels and cell death in U87MG and U251 glioma cells indicating that ROS production is responsible for the DNA breaks and cell death in SFX-01 treated glioma cells.

### 2.5. SFN Induces DNA Laddering and Down-Regulates Autophagic Processes

It has been widely demonstrated that, basally, glioma cells possess appreciable basal levels of LC3-II, suggesting that these cells are able to maintain a significant baseline level of autophagy. Next, we analyzed whether the reduced SFX-01 dependent cell death was associated with increased autophagy and/or induction of apoptosis. As biological markers of autophagy, LC3-I, LC3-II, beclin-1, and p62 were considered. We used (1) the conversion of the dilapidated LC3-I to the lapidated form (LC3-II), in order to verify the activation of autophagosomes; (2) the increase in beclin-1 to better define autophagy; (3) the modulation of p62 for the elimination of autophagosome cargo proteins; and (4) the modulation of Bcl2 and Bax for the inactivation of beclin-1 and induction of apoptosis. The analyses used for this specific aim were a treatment time with SFX-01 of 12–24, as previously indicated, and doses of 5, 10 and 40 µM of SFX-01. Here, we demonstrated that SFX-01 reduced time-dependently the levels of beclin-1 and LC3-II (Figure 6A) as well as the LC3-II/LC3-I ratio being lower in treated cells than the control cells (Figure 6B), whereas the levels of P62 were higher after treatment, suggesting a possible completion of autophagy in 24 h. In Figure 6B, we report the trends of the LC3-II/LC3-I ratio and p62 levels, quantified for U251, U87MG, and A172, and T98G002. The LC3-II/LC3-I ratio was reduced, associated with increased P62 and reduced Bcl2 levels, which were indicative of apoptosis. The kinetics and the role of autophagy in SFX-01 mediated cell death will be better elucidated in a separated and following manuscript when we will analyze the effects of the combination of SFX-01 and radiotherapy. Changes in the DAPI distribution (Figure 6C) suggest the activation of apoptosis. SFX-01-treated cells showed chromatin condensation, an irregular nucleus contour, and pycnotic nuclei. Electrophoretic analyses of U251, U87MG, and A172 DNA extracts showed that the treatment with 5, 10, 20, and 40 µM SFX-01 for 24 h produced a typical distribution of DNA suggestive of apoptosis (i.e., laddering; see Figure 6D). Caspase-3 activation was evident in the Western blotting of the extracts from U87MG and U251, starting from SFX-01 at 5 µM. These data were confirmed by enzymatic assays (Figure 6E). Furthermore, the activities of both caspase-8 and caspase-9 were analyzed.

### 2.6. In Vivo Administration of SFX-01 in U87MG and T98G Subcutaneous Xenografts

To determine the potential therapeutic efficacy of SFX-01 in vivo, two subcutaneous xenograft models with methylated MGMT (U87MG) and non-methylated MGMT (T98G) were used. These cells were injected subcutaneously into athymic female cd1 nu/nu mice (2 tumors/animal; total of 5 animals/group). When the tumors were easily measurable (about 100 mm^3^ in volume), the mice were randomly assigned as controls (vehicle; Group 1) or SFX-01 treated (SFX-01 at 50 mg/Kg/day PO corresponding to 7.7 mg of SFN mg/Kg/day PO; Group 2). Tumor growth was monitored as described in the Materials and Methods section. Overall, SFX-01 alone was well-tolerated, and the animals appeared healthy, with no clinical signs of distress or local/systemic toxicity. In Figure 7, as well as in Table 1, we demonstrate that SFX-01 possessed antitumor ability when used in GBM xenograft models. Panels A and E of Figure 7 show representative images for U87MG xenografts (five harvested tissues). In particular, the reduction of tumor weight was 63.5% (400 mg ± 21 mg vs. 1095 mg ± 111 mg of controls) for U87MG xenografts and 55% (268 mg ± 16 mg vs. 819 mg ± 65 mg of controls) for the T98G model. In order to reduce the likelihood of introducing biases due to differences in tumor engraftment, the tumor progression was analyzed through the time to tumor progression (TTP) parameter (Figure 7C,G and Table 1).

To obtain this parameter, the time necessary for each individual tumor to progress, as defined in the Materials and Methods, was assessed. The incidence of tumor progression over time was plotted using the Kaplan-Meier method, and hazard ratio (HR) values for each comparison were calculated (Figure 7D,H and Table 1). We showed that the TTP in U87MG xenografts was significantly prolonged after SFX-01 treatment vs. controls (21.6 ± 1.0 days vs. 13.70 ± 1.80 days for the control, *p* < 0.005). A similar trend was observed in T98G xenografts. Analysis of the Kaplan-Meyer curves also demonstrated that SFX-01 significantly reduced the tumor progression of U87MG and T98G tumors, with HRs of 4.45 and 2.98 vs. controls, respectively.

### 2.7. Immunohistochemistry Analyses

Excised xenograft tissues from the GBM xenograft mice were examined for the expression of well-characterized tumor markers (Table 2). Representative immunohistochemical images collected from U87MG xenografts are shown in Appendix A. We observed that PAS staining (panels “a and b” for the vehicle and “f and g” for the SFX-01-treated mice) show increased PAS deposits in the latter. These deposits surround little or abnormal vessels and embed close deposits to form a dense and fibrillary structure probably caused by thrombotic vessels pouring their contents into the tumor parenchyma. These structures can be associate to fibrosis which surround both the residual vessels and tumor islets. Immunostaining with α-SMA and CD31 pictured a reduced number of vessels after SFX-01 administration (Table 2 and Appendix A panels “c and d” for vehicle and “h and I” for SFX-01). Xenografts from glioma cell lines show elevated hypoxia as increased tumor growth and abnormal vasculogenesis. So, the evaluation of HIF-1a was used to quantize hypoxia in SFX-01 treated or untreated mice. The administration of SFX-01 reduced significantly HIF1a expression (Table 2 and Appendix A panels “e” for vehicle and “j” for SFX-01). This could be associated with reduced necrosis as shown only in U87MG and not in T98G (Table 2). Reduced tumor growth was associated with statistically significant reduction in Ki67 expression (Table 2 and Appendix A panels “k” for vehicle and “o” for SFX-01). Tunnel positive cells resulted significantly increased in tumor tissues treated with SFX-01 (Table 2). Immuno-histochemical evaluation of ser473 p-Akt expression (Table 2 and Appendix A, panels “m” for vehicle and “q” for SFX-01 administration) indicates a clear inhibition of Akt activation in vivo by SFX-01 administration. Interestingly, p-STAT3, a known in vitro target for sulforaphane activity, was significantly inhibited in vivo, passing from SI = 6 (Table 2 and Appendix A, panel “n”) to SI = 1 (panel “r”). Altogether, the IHC data suggest that in vivo SFX-01 administration maintained the same biological effects obtained in vitro with SFX-01 or SFN. Moreover, as sulforaphane shows inhibitory effects on HDAC, four tissue extracts derived for treatment (s1 to s4) or untreated (v1 to v4) animals were also subjected to enzymatic and western blot analyses for the detection of HDAC activity and HDAC isoenzyme levels. HDAC activity (Table 2) and levels of HDAC1, HDAC2, HDAC4, and HDAC6 (Appendix A were reduced by SFX-01 administration suggesting that SFX-01 is effectively able to realize anti-HDAC activity in vivo.

### 2.8. SFX-01 Increases Disease-Free Survival (DFS) and Overall Survival (OS) in Orthotopic Intra-Brain Tumors

It has previously been described that the bioavailability of SFX-01 is very high for the oral route of administration. Indeed, SFX-01 showed rapid intestinal absorption and production of metabolites, such as cys-SFN, which exhibited high bioavailability and showed better antitumor activities. Hence, the efficacy of SFX-01 in mice receiving luciferase-transfected U87MG cells injected orthotopically into the brain was investigated. As indicated in the Materials and Methods section, we injected a very small number of U87MG cells (3 × 10^3^) into the brain, in order to simulate a recurrence following surgery. In that case, indeed, a relatively small number of cancer cells remains in the operative site. These cells may be able to re-grow and recur.

Similarly, the choice to start the treatment requires the tumor to not be evident in the brain. Therefore, we chose to start the treatment five days after the intracerebral inoculation of tumor cells, as, after five days, no intracranial bioluminescence was detected. We treated the animals for 35 days, as we did with the subcutaneous xenograft experiments. In this case, a follow-up of 125 days without drug administration (a total of 160 days) was considered. The intra-brain tumor growth was monitored by bioluminescence analysis. To monitor the intra-brain tumor growth, we analyzed both the time for the tumor cells to organize and to become visible in the bioluminescence analysis. The parameters used for the analysis were, therefore, the disease-free survival (DFS), defined as the time necessary to exhibit an intra-brain positive bioluminescence signal, and the surrogate overall survival (sOS), defined as the moment at which the mice were euthanized due to the appearance of neurological signs and/or a weight loss of 20% or greater.

Figure 8 shows data for the administration of SFX-01 at 50 mg/Kg/day containing 7.7 mg SFN for 35 days. We demonstrate that the time necessary to detect intra-brain tumors by bioluminescence increased after SFX-01 administration. The controls developed evident bioluminescent lesions between 25 and 40 days after tumor injection (mean values of 31.5 ± 5.3 days), while the SFX-01-treated animals developed bioluminescent lesions after 35–80 days (with a mean of 56.0 ± 14.7 days).

Next, we generated Kaplan-Meier curves for the DFS (Figure 8C). The administration of SFX-01 showed significant protective effects versus CTRL (HR = 3.77, *p* < 0.001). Next, we analyzed the surrogate overall survival (sOS) data. The control mice were sacrificed between 55 and 75 days after the tumor injection, at a mean of 62 ± 8.6 days. SFX-01 significantly increased these values, between 65 to 140 days, with a mean of 99 ± 23.2 days. Next, we generated Kaplan-Meier curves for the sOS (Figure 8E). The administration of SFX-01 showed significant protective effects versus the vehicle (HR = 4.17 and *p* < 0.001).

## 3. Discussion

HGGs comprise a group of primary brain tumors that respond very poorly to conventional therapies, unfortunately typically being diagnosed once patients become symptomatic and when lesions are already very extensive. This makes the surgery, radiotherapy, and chemotherapy-based approaches largely unsuccessful. The combination of these treatments has increased the survival of patients with GBM, but the median survival of patients with GBM remains only about 14.6 months. In recent years, a number of new compounds have been considered as alternative therapeutic tools, to be administered alone or, more effectively, in combination with standards of care. In this report, we analyzed the effects of SFX-01-a fully synthetic and stabilized pharmaceutical product that consists of a complex of alpha-cyclodextrin with synthetic D, L-sulforaphane (SFN)-in glioblastoma preclinical models. The enclosure of SFN in a cloaking ring of alpha-cyclodextrin was an intriguing idea developed in the Evgen Pharmaceuticals. This strategy possesses a great pharmacological potential interest. This alpha cyclodextrin is inert and showed no effectiveness in vitro as previous demonstrated by Leung [43] and Simoes laboratories [39].

Here, we found that SFX-01 significantly reduced the survival of GBM cells and induced apoptosis through the activation of multiple cellular mechanisms. Based on literature data, we first verified whether the in vitro effects mediated by SFX-01 were similar to those obtained using SFN. The used dose ranges were similar between the two compounds. However, the improved bioavailability and stability of SFX-01, with respect to SFN, is indicative of its better potential for in vivo administration in animal models, as well as in clinical settings. Furthermore, this compound is able to cross the blood–brain barrier [59]. These characteristics make it an excellent candidate for the treatment of brain inflammation and brain tumors. Several lines of experimental evidence indicate that SFN is a phytochemical agent that is capable of triggering biologically opposite effects, depending on its concentration and the type of cell considered. Indeed, SFN has shown paradoxical properties, as both a cytoprotective and antitumor agent. Recent reports have indicated that the increased metabolic activity of cancer cells is responsible for the treatment response and recurrence of GBMs.

Basally, the tumor metabolism generates oxygen radicals, which may activate autophagy as a protective mechanism in tumor cells. Working with human cell lines, we have observed measurable LC3-II levels in sub confluent GBM cultures [59,60,61]. Autophagy is a very time-dependent process which can result in pro-death or pro-survival depending on the context and timing of analyses. So, it would be necessary to analyze if SFX-01 increases or reduces autophagy when administrated with radiotherapy. The latter anti-glioma therapy is well known to induce autophagy which can be pro-death in the responder phase and pro-survival in the resistance phase. This it is well elucidated in a manuscript in preparation where the effects of autophagy blockade, by using Bafilomycin A1 and 3-Methyla-adenine, and apoptosis inhibition by caspase inhibitor Z-VAD-FMK are verified in SFX-01-mediated radio-sensitization. We hypothesized that SFX-01 can increase radio-sensitization by repressing RT-mediated pro-survival effects and increasing autophagy [ms in preparation].

The literature has indicated NRF2 (a factor related to nuclear erythroid factor 2), orchestrates the expression of both pro-oxidant and antioxidant components [60] of the NRF2-activated pathway. This enzymatic cascade should lead to the production of phase II detoxifying enzymes. SFN is, indeed, well-known for its ability to reduce ROS accumulation in normal cells (cytoprotective effect) by activating scavenger genes responsible for eliminating free radical products. Paradoxically, however, SFN is also known to increase tumor cell death in different tumor cell systems, through apoptosis induction and the reversion of the autophagic process. The modulation of ROS levels can regulate cell proliferation, cell metabolic capacity, angiogenesis and vasculomimicry [62,63,64]. Although, after the administration of SFX-01, the NRF2 pathways appeared to be activated, intracellular ROS levels were elevated. This could be due to an incomplete attempt to reduce ROS levels. In this regard, the levels of SOD1 and SOD2 were not significantly changed under the treatment with SFX-01, while the levels of HO1 and RAGE were increased. Conversely, Sirt1 levels were reduced after treatment with SFX-01. The effectiveness of SFX-01 was coupled with increased activation of mitochondrial pathways associated with the increased activity of caspases-3, -8, and -9. This suggests a multiple contribution of the intrinsic and extrinsic pathways of apoptosis. High levels lead to cellular stress, promote cell damage by contributing to tumorigenesis. In normal cells there is an equilibrium in ROS levels. In contrast, cancer cells are known to live with increased and persistent oxidative stress due to elevated ROS levels. Elevation of ROS levels, however, induced DNA damage. So, we wanted to verify if SFX-01 triggered this event. We observed, indeed, that SFX-01 triggers the phosphorylation of H2Ax. This it is possible since sulforaphane was able to reduce different kinases [58,61], including phosphatidylinositol 3-kinases (PI3K), DNA-PK (DNA-dependent protein kinase), ATM (ataxia telangiectasia mutated), and ATR (ATM and Rad3-related). The phosphorylation of this histone γH2Ax participates in signaling a DSB. A series of proteins recognize these DSBs and trigger DNA repair, the rate of which is proportional to the sensitivity/resistance to different DNA therapeutics or treatments, such as radiotherapy. SFX-01, by retarding DSB repair through DNA-PKcs inhibition, increases both DNA damage and the percentage of damaged cells that undergo apoptosis.

In addition, we observed reversion of a specialized EMT form involving the perineural/neural cells (PMT [62,63]), indicating the loss of malignancy of established GBM cells and stemness of GICs. Indeed, when we treated the cultures of GICs, we noticed a loss of mesenchymal markers (CD44, CD90, and Stro-1) and a slight increase in neural ones (GFAP and β3-tubulin). This variation was more indicative of the reversion of the malignant phenotype than the differentiation of GICs. From a methodological point of view, it must be emphasized that, in order to detect specific differentiation effects, the growth of GICs is maintained in the absence of serum but in the presence of neurosphere medium. However, it must be considered that the presence of serum in the culture medium alone cannot induce any kind of differentiation in GICs. Unlike in U87MG or U251 GBM cells, the absence of serum can accelerate the induction of the stem cell phenotype. However, we observed that, when SFX-01 was administered to GBM cultures grown in a standard medium containing serum, U87MG and U251 cells mainly triggered a differentiation process leading to the reversion of the EMT phenotype [64]. This is indicative of reduced malignancy of the cells. SFX-01 administration was responsible for cytoskeletal rearrangements, as well as the induction of differentiation, which clearly underlay the defective cell motility. This was an important finding in GBM since, as stated above, its accelerated invasion is often a major feature contributing to the failure of treatment. The acquisition of a less-invasive phenotype has also been associated with reduced vascular properties. Recent experimental evidence has shown that GBM cells and GICs can trans-differentiate into endothelial cells and vascular-type tumor cells. This suggests an alternative angiogenesis mechanism, whereby microvascular circulation is derived from tumor cells through a process known as vasculogenic mimicry (VM) [65,66,67,68,69,70]. Evidence further suggests that this matrix-embedded and perfused microvasculature in the blood plays a vital role in tumor development, regardless of endothelial cell angiogenesis. For example, highly aggressive melanoma cells generate numerous matrix-rich patterned channels containing blood cells [64]; the formation of these channels has been positively correlated with a worse prognosis in patients. Furthermore, we observed that SFX-01 was able to reduce the synthesis and secretion of HIF1α in U87MG and U251 cells in vitro. This could provide a means to reduce the recruitment of GICs in vivo, by reducing the hypoxic/necrotic areas where glioma stem cells develop and grow. Furthermore, the reduced production of HIF1α also leads to the reduced formation of tubules by endothelial cells in vitro. Hence, the vascular properties of GBMs should eventually be reduced.

A series of experimental reports comparing recurrent and non-recurrent human GBMs have demonstrated the activation of STAT3, a well-defined redox-sensitive transcription factor responsible for the aggressiveness of several tumors including GBM. In this study, we demonstrated that SFX-01 was able to inhibit Akt (ser473 and th408 p-Akt isoforms) and Stat3 activation in vivo. SFX-01 reduced tumor growth in subcutaneous xenografts in vivo, independent of the MGMT status. When tested in ototopical intra-brain models, this compound significantly increased the time to appearance of a bioluminescent tumor (disease-free survival), as well as the overall survival of treated mice.

## 4. Materials and Methods

### 4.1. Reagents

A Ki67 antibody was purchased from Dako (Carpenteria, CA, USA) and was used undiluted. Antibodies for anti-active/cleaved caspase-3 (clone E83-77, code ab32042), caspase-9 (code ab2324), SOD1 (code 2770), SOD2 (code 13194), heme oxygenase 1 (clone EP1391Y, code ab52947), RAGE (clone EPR21171, code ab216329), SIRT1 (clone EPR18239, code ab189494), beclin-1 (clone EPR20473, code ab210498), LC3 (clone EPR18709, code ab192890), and SQSTM1/p62 (clone EPR4844, code ab109012) were purchased from Cell Signaling (EuroClone, Milan, Italy). Antibodies against p21 (STJ94862), p27 (STJ94867), cyclin B1 (STJ97479), cyclin D1 (STJ92538), CDK4 (STJ11101180), CD44 (STJ119186), nestin (STJ113858), N-cadherin (STJ94353), catenin-beta (STJ92042), Bax (STJ118092), phospho-Akt (S473) (STJ90166), total Akt (STJ91538), 4E-BP1 (STJ91385), phospho-4E-BP1 (S65) (STJ90778), STAT3 (STJ95808), and phospho-STAT3 (Tyr 705 code STJ90411) were purchased from St John’s Laboratory Ltd. (Docklands Campus University Way, London, UK). Antibodies against p-Rb (clone A-5, code sc-377528), Rb (Rb1, code sc-73598), Oct-3/4 (clone C-10, code: sc-5279), GFAP (clone 2E1 code sc-3673), β3-tubulin (clone 2G10 code sc-80005), GAPDH (clone 0411, code sc-47724), Keap1 (clone G-2, code sc-365626), beta-actin (clone C4, code sc-47778), Nrf2 (clone A-10, code sc-365949), and NFκB p65 (clone F-6, code sc-8008) were purchased from Santa Cruz Biotechnology (Heidelberg, Germany). Secondary antibodies for Western blots were purchased from Santa Cruz Biotechnology. Antibodies were used at dilutions suggested from manufacturers. SFX-01 was provided by Evgen Pharma PLC (Cambridge, UK). SFN was purchased from Selleck chemicals GmbH (Italian distributor, Aurogene, Rome). Cavamax-W6^®^ Pharma Grade is pharmaceutical grade alpha-cyclodextrin (CAS No. 10016-20-3; Empirical formula C36H60O30; Molecular weight 972.84) from Wacker Chemie AG (Adrian, MI, USA).

### 4.2. Cell Lines

Six human glioma cell lines (U251, U118, U373, A172, U87MG, and T98G) were maintained in DMEM containing 10% FBS, 4 mM glutamine, 100 IU/mL penicillin, and 100 μg/mL streptomycin. Short tandem repeat (STR)–PCR profiling demonstrated that the U373MG (Sigma Catalogue number 89081403) and U251 (ECAC catalog 09063001) cells originated from a common cell line [71]; however, these cells showed different biological properties and are models of different evolutionary strains of the same original tumor cell line. It is well known that in vitro subcultures represent, indeed, a selection pressure on cell lines, and, over time, this may result in genetic drift in the cancer cells. Luciferase-transfected U87MG cells were kindly provided by J.E. Heikkila (Abo Akademi University, Turku, Finland). Three patient-derived glioma stem cell lines (BT12M, BT48EF, and BT50EF) were kindly provided by J.G. Cairncross and S. Weiss (University of Calgary, Canada) [72], and one (CSCs-5 [73]) was provided by M. Izquierdo (Universidad Autónoma de Madrid, Spain). Luciferase was inserted into the genome of GSC-5 cells using the pGL4.13 vector (Promega, Milan, Italy) and the jetPEI DNA transfection method (Polyplus, Illkirch, France).

### 4.3. Cell Proliferation Analyses

To determine if SFX-01 modified cell viability, cell cycle progression, and cell death, we analyzed the release of LDH into the medium by using a Cytotoxicity LDH Assay kit-WST (Dojindo Molecular Technologies, Inc., Rockville, MD, USA) and LIVE/DEAD™ Viability/Cytotoxicity kit for mammalian cell analysis (Thermo Fisher Scientific, Inc., Waltham, MA, USA). GBM cells treated with different doses of SFX-01 and untreated control cells were seeded into 24-well plates, at a density of 1 × 10^5^ cells/well at 37 °C for 96 h. The proliferation/cytotoxicity experiments were performed in 5% FBS. The presence of serum guarantees a differentiated status of GBM, through the inhibition of the MES/stem cell phenotype. Cytotoxicity was measured according to the manufacturer’s instructions. We calculated the IC50 values using the Grafit software (Erithacus, Wilmington House, UK). Crystal violet (0.1% in methanol/H_2_O) staining was used to evaluate the antiproliferative effects in short-term (24–72 h) or long-term (15, 21, or 30 days) conditions, according to Cold Spring Harbor Protocols (2016) [74].

GBM cells, treated with different doses of SFX-01 for 48 to 96 h, and untreated controls were subsequently prepared for the detection of cellular apoptosis by using different methodologies such as DAPI staining, TUNEL assays, and caspase-3 detection by Western blotting. Caspase-3, -8, and -9 activities were also evaluated using caspase substrates specifically cleaved by the different caspases (Kaneka Eurogentec SA, Seraing, Belgium), such as Ac-DEVD-pNA (caspase-3 [75]), Ac-IETD-pNA (caspase-8 [76]), or Ac-Leu-Glu-His-Asp-pNA [77] (caspase-9). All the substrates released nitroanilide (NA), which was measured at 450 nm in 96-well plates using a colorimetric microplate reader (Tecan group, Männedorf, Switzerland).

### 4.4. Neurosphere Growth

GICs were grown in a mixture (1:1) of F12 and DMEM (neurosphere medium) containing 30% glucose, B27 (serum-free supplement, 1X), 100 IU/mL penicillin, 100 μg/mL streptomycin, 2 mM glutamine, 10 ng/mL EGF, 50 ng/mL bFGF, and 5 μg/mL heparin. A serum-free medium guarantees an undifferentiated status for GICs [17]. The proliferation of GICs was verified by direct counts of spheres originating from pre-formed small colonies, as well as single cell suspensions (stemness assay) at different times, as previously described [23,78,79] with different pharmacological compounds and treatments. For crystal violet staining, cells were seeded in 6-well plates at a density of 1 × 10^3^ cells per well and allowed to attach to the plate overnight, prior to treatment.

### 4.5. Analysis of Lipid Peroxidation

GBM cells treated with different doses of SFX-01 and untreated control cells were seeded into 96-well microtiter plates, at a density of 2 × 10^4^ cells/well at 37 °C for 48 h. Lipid peroxidation was detected as the amount of malondialdehyde (MDA), in nM/mg of protein of cell extract produced by cells. We used an MDA Assay Kit (Colorimetric from Abcam, code ab118970).

### 4.6. Detection of Intracellular ROS

Reactive oxygen species (ROS) are reactive chemical species containing oxygen. They include peroxides, superoxide, hydroxyl radicals, singlet oxygen, and alpha oxygen. Due to their transient nature, they are easily measured in live cells using fluorescent dye-based assays, such as with DCFDA (Cellular ROS Assay Kit, Abcam number ab113851). The intracellular levels of ROS were detected using a fluorescence microplate reader (Labsystems Fluorskan Ascent 374) from ThermoFisher Scientific (Life Technologies Italia, Monza, Italy).

### 4.7. Cell Lysate, Western Blot, and Akt/mTOR Enzymatic Analyses

Following treatments, cells grown in 90 mm-diameter Petri dishes were washed with cold PBS and immediately lysed with 1 mL of lysis buffer containing a proteinase and phosphatase inhibitor cocktail. The total lysates were electrophoresed in 10% SDS-PAGE gels, and the separated proteins were transferred to a nitrocellulose membrane, which was probed with the appropriate antibodies using the conditions recommended by the suppliers. Protein expression was normalized using anti-GAPDH or β-actin antibodies. For the Akt/mTor enzymatic assay, we used the In-Cell ELISA (also known as cell-based ELISA), an immune-cytochemistry method used to quantify target proteins or post-translational modifications of a target protein directly in cultured cells. We tested the expression of p-Ser473 Akt, p-Thr408 Akt, p-Ser2448 mTOR, Thr46/47-4E-BP1, and pSer235/236-S6 by measuring the optical density at 450 nm with a TECAN Elisa reader (Tecan Italia, Cernusco sul Naviglio, Italy), according to the In-Cell ELISA protocol from Abcam [80], as previously described [80]. Cells were plated in 96-well multi-plate at 5000 cells/well. After adhesion, complete medium, supplemented or not according to the experimental conditions, was added, and the test was performed at 3 h. No detection of cell number was analyzed, as the analysis determined whether no cell death had occurred within a short time.

### 4.8. Immunofluorescence (IF) Studies

GBM cell lines and GICs were used for IF analyses. GBM cells and GIC spheres were seeded on glass coverslips pre-treated with poly-L lysine (30 µg/mL). After cell adhesion (usually 1 h), cells were incubated with the respective media establishing the experimental conditions for an additional 24 h. Next, the slides were fixed with 4% paraformaldehyde for 20 min, permeabilized with 0.3% Triton-X-100 (Sigma Aldrich, St. Louis, MO, USA) for 5 min, and, finally, incubated with the following primary antibodies (according to their data sheets): anti-OCT3/4, Ki67, β3-tubulin, NFH, GFAP, Sox2, Stro-1, CD90, and CD44. Fixation and washing were performed at room temperature, while the incubation with primary antibody was carried out overnight. After three washes with Ca^2+^ and Mg^2+^ phosphate-buffered solution (PBS), AlexaFluor 488 anti-rabbit IgG, AlexaFluor 595 anti-goat IgG, or AlexaFluor 633 anti-mouse IgG secondary antibodies were added for a further 30 min. The slides were washed three times and mounted in DAPI (0.5 μg/mL) containing Vectashield Mounting Medium (Molecular Probes, Invitrogen, Carlsbad, CA, USA). Adherent GBM cells were also seeded at 1 × 10^4^ cells/well on 8-well chamber slides (Fisher) and incubated as described above. Slides for coverslips and chamber slides were examined using a FLoid™ Cell Imaging Station (Thermo Fisher Scientific, Waltham, MA, USA) or confocal microscope (Leica TCS SP5, Mannheim, Germany).

### 4.9. FACS Analyses

The expression of antigens on cells untreated or treated with different doses of SFX-01 was quantified by flow cytometry with paraformaldehyde-fixed and permeabilized cells. Washed cells were incubated for 1 h at 4 °C with selected primary antibodies, followed by CY5-conjugated anti-rabbit IgG H&L or PE-conjugated anti-mouse IgG (purchased from Abcam, Cambridge, UK) for an additional 30 min. All the samples were analyzed by using a BD Accuri™ C6 Plus Flow cytometer (Becton Dickinson Italia Spa, Milan, Italy), equipped with a blue laser (488 nm) and a red laser (640 nm). At least 10,000 events were acquired.

### 4.10. Migration and Invasion Assays

The test of migration and invasion was previously described [81]. Briefly, cell migration assay was performed using a 48-well Boyden chamber (Neuroprobe, Inc, Gaithersburg, MD, USA) containing 8-μm polycarbonate filters (Nucleopore, Concorezzo, Italy). Filters were coated on one side with 50 μg/mL laminin, rinsed once with PBS, and then placed in contact with the lower chamber containing DMEM. Untreated or SFX-01 treated GBM cells were added (75,000 cells/50 μL) to the top of each chamber and allowed to migrate through coated filters for 6 h. In the lower compartment of the invasion chamber 5% FBS-containing medium was added as a chemo-attractant. Invasion assay was performed in invasion chambers containing a membrane coated with Matrigel™. The digestion of Matrigel allowed the migration of cancer cells. At the end of the incubation, the migrated/invaded cells attached to the lower membrane surfaces were fixed, stained with Diff quik (MBT srl (Treviso, Italy) and counted at a ×40 magnification by standard optical microscopy. The results of three separate experiments of migration and invasion are presented as the mean ± SD. Each experimental group consisted of 5 samples.

### 4.11. Conditioned Medium Preparation, Vasculogenic Mimicry, and Endothelial Tubule Formation

U251 and U87MG cells were cultured for at least three passages under controlled conditions (37 °C and 5% CO_2_). Then, they were passaged in T75 flasks (10,000 cells/cm^2^). After attachment overnight, these cells were maintained for 24 h of incubation in serum-free medium and then collected and centrifuged at 260× *g* at 4 °C for 10 min, and the supernatant was kept at −80 °C until required for the experiments. Human Umbilical Vein Endothelial Cells (HUVEC) catalog number C0035C were maintained in endothelial cell medium Catalog number M2005 were purchased from ThermoFisher Scientific, Monza, Italy (Monza, Italy). The tube formation assay [82] was carried out using an in vitro Matrigel assay kit (Chemicon, Millipore), following the manufacturer’s instructions, by coating 15-well micro-slides (10 μL/well) with IBIDI (Munich, Germany). The method was applied for both the vasculogenic mimicry (VM [83]) and angiogenic analyses using GBM cell lines (CTRL and treated with SFX.01) and endothelial cells (CTR and conditioned cells). In brief, the Matrigel was solidified at 37 °C for 30 min, and 15,000 HUVECs/well or GBM cells were added to chilled pellets and incubated with EGM-2 or DMEM, respectively, in the presence of conditioned medium (CM) derived from glioblastoma cell lines (1:4) for up to 12–16 h or SFX-01 at different concentrations. The degree of the angiogenic and vascular (VM) response was assessed at the end of treatment, using an inverted phase contrast microscope, by evaluating the branching index. Each well was photographed, and the relevant acquired images were analyzed using the ImageJ analysis software. The mean values and standard deviations (SDs) for the vessel counts were determined for each analysis.

### 4.12. Animal Experiments: Subcutaneous Xenograft Model

After 1 week of quarantine, female CD1-nu/nu mice at 6 weeks of age (purchased from Charles River, Milan, Italy), following the guidelines established by our Institution (University of L’Aquila, Medical School and Science and Technology School Board Regulations, complying with the Italian government regulation n.116 27 January 1992, for the use of laboratory animals, code 555/2017-PR (CE5C5 01-4-2017 approved data 7.7.2017), received two subcutaneous flank injections of 1 × 10^6^ U87MG and T98G cells. Next, the animals were randomly divided into two different groups, containing 5 animals/group (10 subcutaneous tumors each with a volume of 0.8–1.3 cm^3^), as follows: Group I, Control (CTRL, vehicle, 200 μL of PBS containing 0.9% NaCl, by oral gavage), and Group II, SFX-01 (50 mg/Kg/day, 5 days each week, corresponding to 7.7 mg/Kg SFN, by oral gavage; SFX-01 was dissolved in 200 μL of PBS containing 0.9% NaCl). Tumor growth was assessed biweekly by measuring the tumor diameters with a Vernier caliper using the formula TW (mg) = tumor volume (mm^3^) = 4/3πR1xR2xR3, in which R1, R2, and R3 are the half-diameters (rays)-the shorter diameter is the thickness/height of the tumor, while the larger diameters are the length and width of the tumor [54]. In order to reduce the probability of bias due to differences in tumor engraftment, tumor progression was analyzed through the time to progression (TTP), defined as the time (in days) necessary for the tumor to double in volume. The percentage of animals with tumor progression was plotted (with respect to time) by using the Kaplan–Meier distribution, as previously described [83,84,85,86].

### 4.13. HDAC Activity in Subcutaneously Xenografted Tumors

The histone deacetylase activity in tissue extracts from U87MG and T98G xenografts of the control and SFX-01 treatment groups was measured using the HDAC Activity Colorimetric Assay Kit from Biovision Inc. (Italian distributor DBA srl, Segrate, Milan, Italy), as previously described [85].

### 4.14. Immunohistochemical Analyses

Xenograft tissues were collected after the euthanasia of the animals at the end of the experiments. The tissues were transferred into a 4% formalin solution in PBS and processed for paraffin embedding using standard protocols. Indirect immuno-peroxidase staining was performed on 4 μm paraffin-embedded tissue sections. We analyzed several markers via the streptavidin-avidin staining protocol and counterstained with Mayer hematoxylin solution, except for the TUNEL staining, in which the counterstaining could mask nuclear positivity. A consensus judgment, as indicated in our previous report [86], was adopted as the proper immunohistochemical score for the tumors, based on the strength of staining: negative (score 0), weak (score 1), moderate (score 2), or strong (score 3). In each category, the percentage of positive cells was assessed by scoring at least 1000 cells in the area with the highest density of antigen-positive cells. The cytoplasmic/membrane staining intensity was graded as follows: 0 = negative; 1 = less than 10% of positive cells; 2 = positive cells in the range of 10–50%; and 3 = more than 50% positive cells. The overall expression was defined by the staining index (SI) and ranged between 0 and 9, with an SI ≤ 4 indicating a low expression. Tumor micro vessels were counted at 400× magnification in five arbitrarily selected fields, and the data were presented as number of CD31+ mouse micro vessels per 100× microscopic field for each group. The Ki67 labeling index was determined by counting 500 cells at 100× magnification and determining the percentage of cells positively stained for Ki67. Apoptosis was measured as the percentage of TUNEL-positive cells measured in five random fields (400×) using a TACS Blue Label kit (R&D Systems, Inc., Minneapolis, MN, USA).

### 4.15. Orthotopic Intracranial Model

Female CD1 nu/nu mice were inoculated intracerebrally, as previously described [85,86], with luciferase-transfected U87MG and GSCs-5 cells. Five days after inoculation, when no luciferase activity was intracranially detectable, the animals were randomized with 10 mice per group, as follows: Group I, Control (CTRL, vehicle, 200 µL of PBS containing 0.9% NaCl, by oral gavage), and Group II, SFX-01 (50 mg/Kg/day, 5 days each week corresponding to 7.7 mg/Kg SFN, by oral gavage; SFX-01 was dissolved in 200 µL of PBS containing 0.9% NaCl). Tumor growth was assessed every 5th day by the bioluminescence intensity (BLI) using the UVITEC Cambridge Mini HD6 (UVItec Limited, Cambridge, United Kingdom). The animals were anesthetized, and luciferin (150 mg/kg) was injected intraperitoneally 15 min prior to imaging. To simulate the tumor burden of the clinical condition, usually determined in terms of the complete GBM resection pathological condition for the time at which surgery occurs (a low number of tumor cells in wound bed causing regrowth and recurrence), we inoculated a small number of cells (3 × 10^3^), and SFX-01 administration was started 5 days after intracranial tumor injection, before bioluminescence was detected. The surrogate overall survival (sOS) was defined as the moment at which the mice were euthanized due to the appearance of neurological signs and/or a weight loss of 20% or greater. The disease-free survival (DFS) was defined as the time necessary to exhibit an intra-brain positive bioluminescence signal. The mice were euthanized when they displayed neurological signs or weight loss of 20% or greater. The sOS values were recorded.

### 4.16. Statistics

Numeric data are expressed as means ± SDs or medians with 95% confidence intervals (CIs). ANOVA was followed by Tukey’s test. The incidence over time of the tumor progression of subcutaneously xenografted tumors, the disease-free survival (DFS), and the OS were analyzed using Kaplan–Meier curves and Gehan’s generalized Wilcoxon test. *p* values < 0.05 were considered statistically significant. The MedCalc statistical analysis software package (version 10.0) was used for statistical analysis and graphic presentation.

## 5. Conclusions

In this work, we used in vitro and in vivo experimental models to assess the therapeutic potential of SFX-01 for GBM treatment. Our data suggest that SFN targets ER stress and NRF2, paying as a “master regulator” of the body’s stress response. Secondary but very important effects are the inhibition of HDAC activity especially on class I HDACs and inhibition of STAT3 with reduced activity of NFKB pathways which modulate progression of GBM and induce ROS. So, metabolic, oxidative, and epigenetic modifications are the bases of the different effects of SFN. Our study provided detailed evidence that SFX-01 induced cell death and inhibited the growth of GBM cells and xenografts through the modulation of multiple cell-signaling pathways, demonstrating its potential as a new compound for GBM treatment. SFX-01 was also highly effective in eliminating GSCs through the induction of cell differentiation, compared to the less-malignant astrocyte phenotype. SFX-01 also reduced the vasculogenic properties of GBM cells, thus inhibiting the growth of GBM xenografts in vivo. SFX-01 triggered DNA damage and increased RT-mediated γ-H2Ax expression through the production of ROS. Overall, our preclinical data indicated that the effects of SFX-01 are due to multiple molecular mechanisms, participating in switching off autophagy and switching on apoptosis, which are responsible for SFX-01-mediated cell death.

## Figures and Tables

**Figure 1 pharmaceuticals-14-01082-f001:**
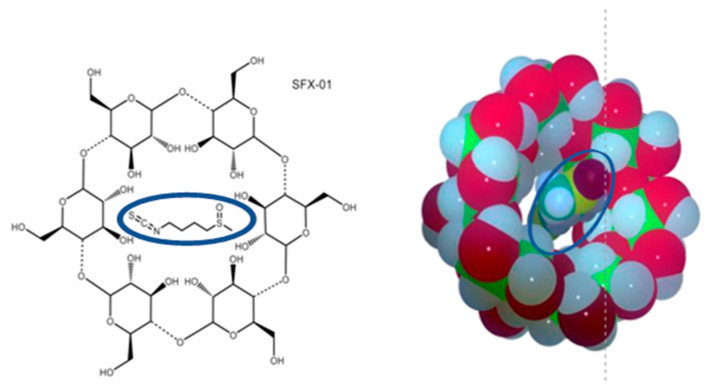
Chemical structure and three-dimensional representation of SFX-01. The cyclodextrin alpha (Cavamax-W6^®^ (Wacher Chemie AG, Adrian, MI, USA) shows 6 glucose units has the smallest cavity of the parent cyclodextrins. It is useful for solubilizing, stabilizing, or delivering small molecules. SFN structure is present in the core of molecule (outlined with a blue line).

**Figure 2 pharmaceuticals-14-01082-f002:**
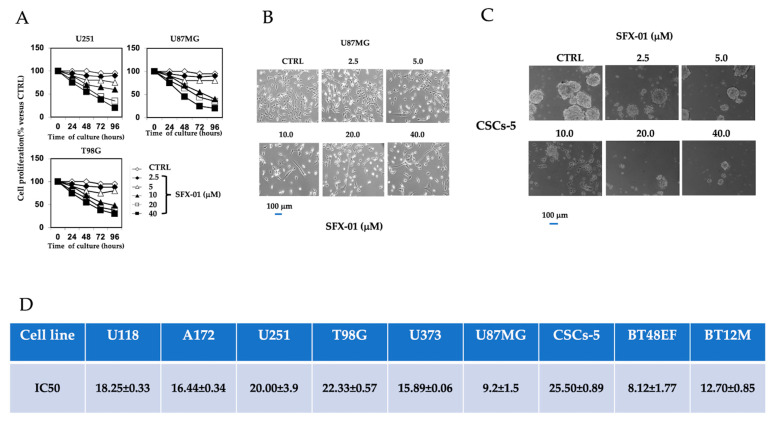
SFX-01 inhibited cell proliferation of GBM cell lines and GICs: (**A**) Growth rates at 24, 48, 72, and 96 h measured in U251, U87MG, and T98G cultured with different doses of SFX-01 (combined experiments for time and dose dependence). (**B**) Representative images of morphological changes caused by SFX-01 in U87MG cells treated for 48 h. SFX-01 caused the acquisition of cell extensions probably associated with the type I astrocytic phenotype. (**C**) Representative images of pre-formed GSC-5 neurospheres treated with different doses of SFX-01. The bar represents 50 µm. Cells were maintained and cultured for the experiments in serum-free DMEM/F12 medium mixture (1:1). (**D**) IC50 values (mean values +/− standard deviation (SD) calculated by plotting the percentage of cells/colonies, normalized versus untreated controls, against each concentration of SFX-01 used as specified in the Materials and Methods section.

**Figure 3 pharmaceuticals-14-01082-f003:**
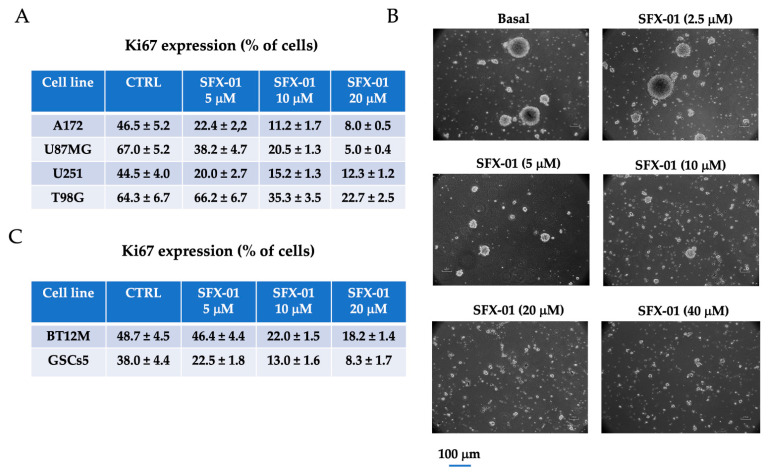
SFX-01 inhibits cell proliferation in GBM cell lines and GICs: (**A**) Ki67-positive cells-percentage of Ki67-positive cells in A172, U87MG, U251, and T98G cell lines treated with 5, 10, and 20 µM SFX-01. (**B**) Sphere formation assay performed on BT12M treated with different doses of SFX-01. Bars represent 50 µm. (**C**) Ki67-positive cells in BT12M and GSC-5 GICs treated with 5, 10, and 20 µM SFX-01.

**Figure 4 pharmaceuticals-14-01082-f004:**
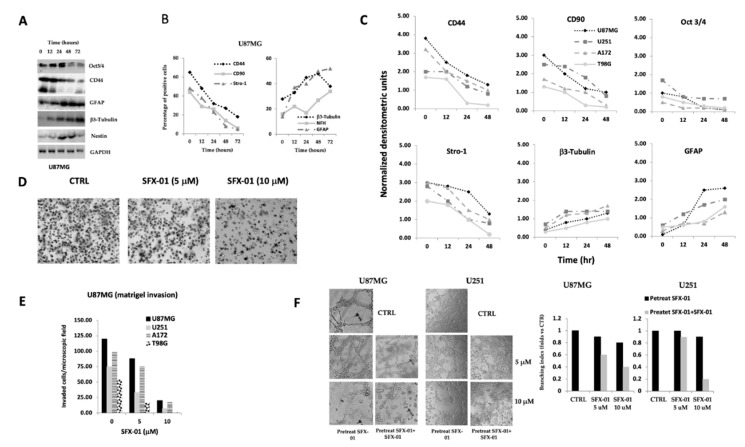
SFX-01 reduces PNT through the reduction of mesenchyme markers and induction of differentiated ones. (**A**) Representative western blots performed on U87MG cell extracts: Time-dependent modulation of CD44, Oct3/4 (mesenchymal/stem cell markers), β3-tubulin, nestin, and GFAP (differentiated markers). (**B**) Representative FACS analysis: Time-dependent modulation of CD44, Stro1, and CD90 as MES/stem cell markers; β3-tubulin and neurofilaments (NF200) as neural markers; and GFAP as an astrocyte marker. (**C**) semi-quantitative analyses (normalized densitometric values obtained by ImageJ in western blots performed on cell extracts from U87MG, U251, A172 and T98G glioma cells. (**D**) matrigel invasion assay (representative pictures from U87MG cells) and (**E**) counts of invaded cells/microscopic field in U87MG, U251, A172 and T98G cells treated with 5 and 10 µM SFX-01 (**F**) Analysis of vasculomimicry (VM) of U87MG (and U251, for comparison of more-differentiated cells) showed reduced Matrigel invasion by cells treated with 5 or 10 µM SFX-01. Pre-treatment with SFX-01 decreased VM activity. Quantification of VM was performed by the evaluation of tubule structure and branching index calculation.

**Figure 5 pharmaceuticals-14-01082-f005:**
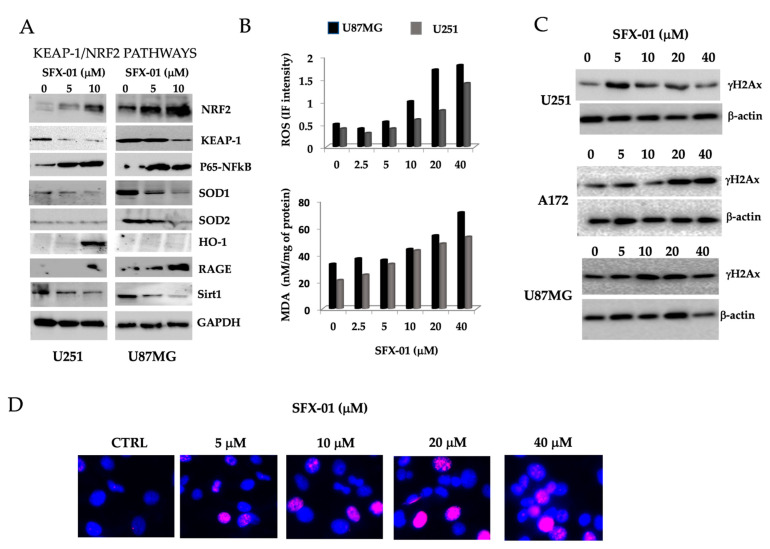
Sulforaphane induces reactive oxygen species (ROS) and amplified the RT-mediated oxidative nature of the microenvironment: (**A**) NRF2/Keap1 pathway activity in SFX-01-treated U251 and U87MG cultures growing with different doses of SFX-01. By Western blotting, we analyzed the expression of NRF2, KEAP1, P65-NFkB, SOD1, SOD2, heme oxygenase-1 (HO-1), RAGE, and Sirt1. (**B**) ROS levels (measured according to 2′,7′-dichlorofluorescin diacetate, DCFDA) and lipid peroxidation (measured according to malondialdehyde peroxidation, MDA). Differences were statistically significant with *p* < 0.005 since SD was <15% of mean value (**C**) Western blotting assays performed in U87MG, U251, and A172 cell extracts for γ-H2Ax expression after treatments with increasing doses of SFX-01. (**D**) Immunofluorescence analysis of γ-H2Ax foci in U87MG cells treated with 5, 10, 20, and 40 µM SFX-01. Fixed cells were analyzed for expression of positive nuclei and, when possible, for number of γ-H2Ax foci.

**Figure 6 pharmaceuticals-14-01082-f006:**
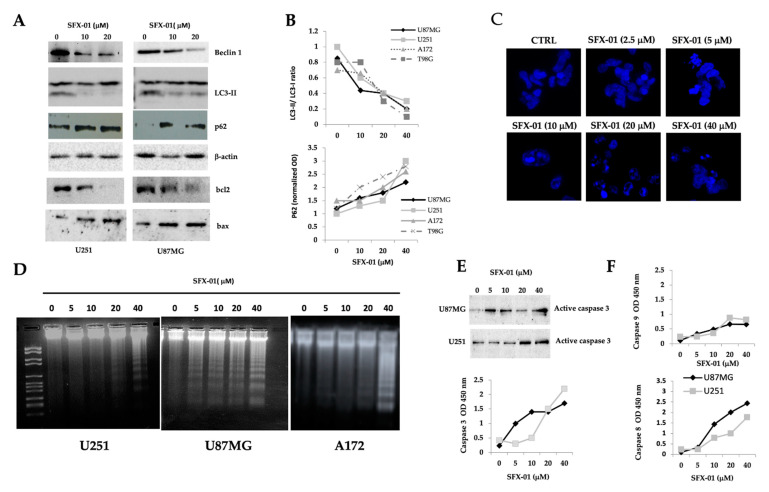
SFX-01 reduced basal autophagy and induced apoptosis in GBM cell lines: (**A**) Beclin-1, LC3, P62, Bcl2, and Bax expression according to Western blots, performed on U87Mg and U251 cell extracts. A total of 40 µg of proteins per lane were loaded. To quantify autophagy, Western blots were repeated three times and each band was analyzed using the ImageJ software (v. 1.8.0). LC3-II and LC3-I bands were normalized to β-actin before calculation of LC3-II and LC3-I ratio values. (**B**) LC3-II/LC3-I ratios determined from Western blots for U87MG, U251, T98G, and A172 cells. (**C**) Morphological changes in U87MG nuclei stained with DAPI for control cells and cells treated with different SFX-01 doses. (**D**) DNA laddering performed for DNA extracted from U251, U87MG, and A172 cell lines, control and treated with 5, 10, 20, and 40 µM SFX-01. (**E**) Western blots and enzymatic assay performed on U87MG and U251 cell lysates showing caspase-3 activation (cleaved caspase-3 identification), as well as quantification of caspase-3 activity by EIA assay for caspase-3. Differences were statistically significant with *p* < 0.005 since SD was <15% of mean value (F) Quantification of activity of caspase-8 and caspase-9 by EIA determinations. Differences were statistically significant with *p* < 0.005 since SD was <15% of mean value.

**Figure 7 pharmaceuticals-14-01082-f007:**
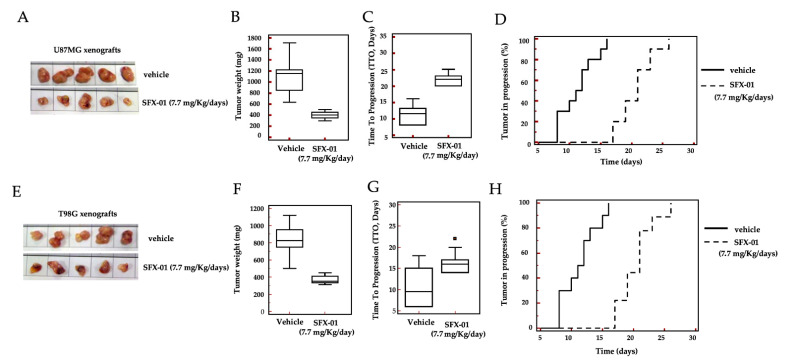
In vivo experiments: SFX-01-mediated tumor growth reduction and radio-sensitization of U87MG and T98G xenografts. Each experimental group is represented by five mice with two tumors in the flank: (**A**) Five representative tumors harvested from U87MG xenografts (control and after treatments). (**B**) Quantification of tumor weights in 10 tumors harvested from treated and untreated animals. Statistical analysis is shown in Table 1. (**C**) Tumor to progression (TTP) analysis. (**D**) Percentage of mice in progression plotted for the time (Kaplan-Meier curves). Analysis of Kaplan-Meier curves demonstrated that SFX-01 reduced the tumor progression rate of U87MG. (**E**) Five representative tumors harvested from T98G xenografts (control and after treatments). (**F**) Quantification of tumor weight in 10 tumors harvested from treated and untreated animals. Statistical analysis is shown in Table 1. (**G**) Tumor to progression (TTP) analysis. (**H**) Percentage of mice in progression plotted for the time (Kaplan-Meier curves). Analysis of Kaplan-Meier curves demonstrated that SFX-01 reduced the tumor progression rate of T98G. The dose equivalent of 7.7 mg/Kg contained in 50 mg/Kg SFX-01 is indicated in the panels of this figure. Statistics are indicated in the Appendix A.

**Figure 8 pharmaceuticals-14-01082-f008:**
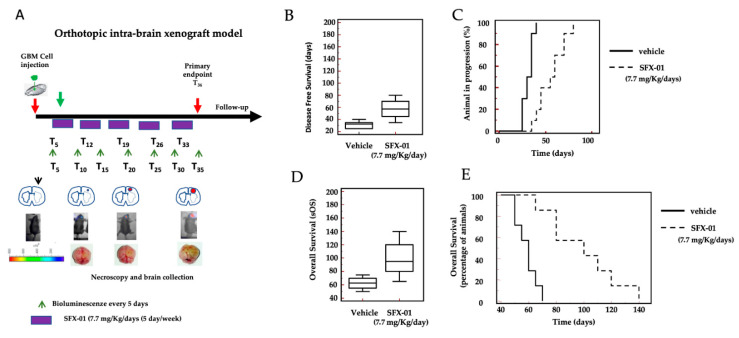
SFX-01 inhibited tumor growth in an orthotopic model for GBM in mice. U87MG cells expressing luciferase were orthotopically implanted into athymic nude mice, and tumor growth was monitored using a UVITEC Cambridge Mini HD6 (UVItec Limited, Cambridge, UK) imaging system for the detection of bioluminescence. Bioluminescent signals were measured every week, in order to record the time of bioluminescence signal appearance (equivalent of disease-free survival, DFS): (**A**) treatment scheme; (**B**) disease-free survival (days) graph; (**C**) representative Kaplan-Meyer analysis for DFS; (**D**) surrogate overall survival (sOS, days) graph; and (**E**) Kaplan-Meier analysis for sOS. The data presented here indicate that SFX-01 increased both the DFS and sOS of animals bearing orthotopic intra-brain U87MG tumors. The dose equivalent of 7.7 mg/Kg contained in 50 mg/Kg SFX-01 is indicated in the panels of this figure.

**Table 1 pharmaceuticals-14-01082-t001:** Statistical analyses of tumor weight and Time to Progression in U87MG and T98G xenografts.

U87MG							
Groups	Tumor Weight (mg ± SE)	Statistics	TTP (Days)	Statistics	HR	CI 95%	Statistics
1. CTRL	1095 ± 111		13.7 ± 1.8				
2. SFX-01	400 ± 21	*p* < 0.001	21.6 ± 1.0	*p* < 0.001	4.45	1.51 to 15.22	*p* < 0.0001
T98G							
3. CTRL	819 ± 65		13.2 ± 0.7				
4. SFX-01	368 ±16	*p* < 0.0001	17.0 ± 0.5	*p* < 0.001	2.98	1.09 to 8.20	*p* < 0.001

**Table 2 pharmaceuticals-14-01082-t002:** Immuno-histochemical and enzymatic evaluation on markers of proliferation, apoptosis, autophagy, angiogenesis, stemness and signaling in untreated (control) and treated with SFX-01 in U87MG and T98G xenografts.

Groups	Vessel Count (CD31)	Ki67 Percentage	Apoptosis (% of Tunel Positive Cells)	Necrosis (% of Necrotic Area)	HIF1α (Arbitrary Score)	Phospho-(ser473)-Akt (Arbitrary Score)	HDAC Activity OD_405nm_/µg Protein	p-Stat3 (Arbitrary Score)
U87MG Xenografts								
1. CTRL	30.8 ± 5.0	35.5 ± 4.0	<2	15.0 ± 5.5	6 (3 × 2)	9 (3 × 3)	1.687 ± 0.244	6 (2 × 3)
2. SFX-01 (50 mg/Kg/day)	18.0 ± 4.0 *	21.5 ± 0.5 **	15.5 ± 5.5 **	10.0 ± 5.0	2 (2 × 1)	2 (1 × 2)	0.557 ± 0.188	1 (1 × 1)
T98G Xenografts								
3. CTRL	22.2 ± 2.4	20.7 ± 4.0	<2	8.0 ± 4.0	4 (2 × 2)	6 (2 × 3)	1.332 ±0.330	6 (2 × 3)
4. SFX-01 (50 mg/Kg/day)	17.0 ± 2.0 *	8.0 ± 2.5	8.5 ± 2.0	15.0 ± 3.5	2 (1 × 2)	1 (1 × 1)	0.233 ± 0.088	1 (1 × 1)

Arbitrary score (staining index-SI 0–16): intensity (score 1) × percentage of positive cells (score 2). SI < 4 low expression. * *p* < 0.01; ** *p* < 0.005.

## Data Availability

The data are contained within the article or Appendix A.

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
