# Peer review of "Multiple Antitumor Molecular Mechanisms Are Activated by a Fully Synthetic and Stabilized Pharmaceutical Product Delivering the Active Compound Sulforaphane (SFX-01) in Preclinical Model of Human Glioblastoma"

_pharmaceuticals, 2021, doi:10.3390/ph14111082_

Round 1
Reviewer 1 Report
Re ethics concern, see my comment below re NCT02970682.
This paper has been improved but is still too sloppily written to be acceptable. They require a professional medical editor who is familiar with the basics of cancer research. The fundamental idea of enclosing sulforafene in a cloaking ring of alpha-cyclodextrin is intreguing and of great potential interest.
My main objection to the technical part of what the authors did is the lack of controls given alpha-cyclodextrin alone. This must be done before publication. If this was done and I misunderstood the paper, my appologies, but make the text clearer please in that case.
L35. would it be correct to say “reversal of EMT”
L43, grammar error.
Run-on paragraphs remain. I suggest breaking up the larger ones. Examples, at lines 65, 73, 81.
Line 89, is misleading. Extreme driver heterogeneity exists even within each subgroup.
Line 91, This statement may remain, but authors please note my disagreement with that view.
Lines 97 thru 99 requires rewording… “ As opposed to what, HGGs contain elevated percentages of cells, resulting in a transformed, self-maintaining, proliferating, multipotent, and stem cell phenotype (GSCs).” What are they saying ?
Did the authors intend to say “As opposed to that GBM contain elevated percentages of malignant cells that are self-maintaining, proliferating, multipotent, and of stem cell phenotype.” ?
If so, say so simply. If the authors wish to say something like that then they must also say how their use of GIC differs from the common use of GSC, the subset of cells within a GB that have stem attributes and stem cell markers. This matter of stemness requires its own paragraph. It must be explicitly stated that stemness refers to a wide range of attributes, some of which are quite unrelated to others. Also it must be mentioned that eliminating one of these attributes may well leave the other stem attributes unaltered.
“These cells are called glioma-initiating cells “ must start a new paragraph. After any discussion of stem cell “markers” explicit mention that “markers” are usually also mediators - creators - of one or another stem attribute.
Re lines 101, 102, “This cell population develops in highly hypoxic areas of a tumor, where typical palisading necrosis with elevated inflammation is present [13-16].” Would it not be more correct that these hypoxic areas are enriched for stem attributes but these GB stem cells can be found also in non hypoxic areas ? in this context it is crucial to mention that interconversion stem to nonstem and nonstem to stem has been well documented in GB.
It must be mentioned that sulforaphene is commonly available over-the-counter, without a prescription, in many jurisdictions. It is also popularly used by cancer patients.
Figure 01 must be color coded or otherwise made clear to indicate which is sulforaphene and what is alpha-cyclodextrin. This is obvous to me [us] but one cannot assume this will be for others. The legend must be appropriately amended.
Line 113, again use of word “modulates” leaves readers unclear re. increase or decrease ?
Why not say simply what you intend ?
Line 116, you cannot go back and forth between GB and then start lumping them together with “brain tumors”. Are the authors suddenly talking about CNS lymphomas and brain metastases from breast cancer too ?
Line 129, what are “Basal GBMs…” ?
NCT02970682 ended 3 years ago. I never saw results published. Why not ? if the authors mention NCT02970682 they must either report results or state clearly why they are not stating the results. My apologies if I missed the results but I was unable to find these. In any case results must be mentioned.
I will be glad to finish reviewing after the above deficiencies have been remedied or an explanation why I am in error has been provided.
Reviewer 2 Report
Since this revised version is a split report from a previous submission, I will treat this as a new manuscript and focus my review on the present MS.
Figure 1: Why do sub-figures A to C have a border and D and E don’t, please harmonize. Also, I would advise to summarize the calculated IC50-values in a separate table for improved clarity.
Line 465ff: Please provide references for the process of PMT
Line 481: data not shown is uncommon for open access journals with no size restrictions. Please add these data as supplementary data.
Figure 4: I don’t fully agree with the conclusions drawn from these data. 1. I would interpret the first two sub-figures as reduction in stemness and 2. The last two sub-figures as preventing vasculogenic mimicry and not PMT. Therefore I advise ro re-phrase the related paragraphs. In addition, it would be more interesting to analyze the stemness marker in an actual stem-like line (or multiple preferably) as this would mechanistically explain the findings presented in Fig. 3.
Line 561: similar to line 481
Figure 5 and related text: Is it really contrasting that detoxifyingn enzymes are increased when ROS are present in the cells? I would rather argue that this is a compensatory mechanism towards ROS-induction. Thus, I would believe that at early timepoint ROS could be measured, but the response (protein expression), would still be low. This can also easily answered experimentally. In addition, it would be interesting to see if ROS-production really leads to DNA damage and decreased proliferation. This as well can be easily tested by applying a ROS-scavenger and testing if the cells remain insensitive.
Line 584: “data not shown”
Figure 6: The analysis of the autophagic pathway doesn’t allow for many conclusions. 1: Autophagy is a very time-dependent process and it is advisable to analyze multiple timepoints to get an impression of the kinetics. 2. Autophagy can pro-death or pro-survival depending on the context and as such it would be interesting (and essential) to analyze if blockage of autophagy (e.g. BafA1 and/or 3-MA) can prevent or enhance the proliferation-inhibiting properties of the drugs. As the analyses stands currently, it is of little use.
Figure 7: An arrow/timeline depicting the treatment scheme would be much appreciated (see Fig. 8). Also, I assume that tumor growth was monitored throughout the experiment. As such, it is advisable to present to changes in tumor value over time and not only if the tumor progresses, because generally subcutaneous tumor display a continuous growth.
Table 2: Please provide representative data for each analyzed parameter as supplements.
Minor points:
Line 43: angiogenetic --> change to angiogenic
Please make sure to put to figures closer to the respective text fragment, especially figure 7 is quite far away from the text.
Line 770: RT is not the only therapeutic option. Re-Phrase
